# Etrolizumab-s fails to control E-Cadherin-dependent co-stimulation of highly activated cytotoxic T cells

Maximilian Wiendl[1], Mark Dedden[1], Li-Juan Liu[1], Anna Schweda[1], Eva-Maria Paap[1], Karen A.-M. Ullrich [1], Leonie Hartmann[1], Luisa Wieser[1], Francesco Vitali[1], Imke Atreya[1,2], Tanja M. Müller[1,2], Claudia Günther[1,2], Raja Atreya [1,2], Markus F. Neurath [1,2] & Sebastian Zundler [1,2] ✉

Despite promising preclinical and earlier clinical data, a recent phase III trial on the anti-β7 integrin antibody etrolizumab in Crohn's disease (CD) did not reach its primary endpoint. The mechanisms leading to this outcome are not well understood. Here we characterize the β7+ T cell compartment from patients with CD in comparison to cells from individuals without inflammatory bowel disease. By flow cytometric, transcriptomic and functional profiling of circulating T cells, we find that triple-integrin-expressing (α4+β7+β1hi) T cells have the potential to home to the gut despite α4β7 blockade and have a specific cytotoxic signature. A subset of triple-integrin-expressing cells readily acquires αE expression and could be co-stimulated via E-Cadherin-αEβ7 interactions in vitro. Etrolizumab-s fails to block such αEβ7 signalling at high levels of T cell stimulation. Consistently, in CD patients treated with etrolizumab, T cell activation correlates with cytotoxic signatures. Collectively, our findings might add one important piece to the puzzle to explain phase III trial results with etrolizumab, while they also highlight that αEβ7 remains an interesting target for future therapeutic approaches in inflammatory bowel disease.

Immune cell infiltration to the gut is a hallmark of the dysregulated intestinal immune response in inflammatory bowel diseases (IBD) such as Crohn's disease (CD) and ulcerative colitis (UC)[1–3]. Consistently, one successful therapeutic approach is the inhibition of intestinal cell trafficking pathways with drugs such as vedolizumab, a monoclonal antibody targeting the gut homing integrin α4β7[4,5], or ozanimod, a small molecule agonist of sphingosine 1 phosphate receptors[6].

Importantly, cell trafficking circuits in the gut are marked by an organ-specific regulation[7]. On the one hand, α4β7 integrin is specifically induced on T cells primed in the gut-associated lymphoid tissue[8,9]. On the other hand, the α4β7 ligand mucosal addressin cell adhesion molecule (MAdCAM)-1 is quite specifically expressed on the intestinal endothelium[10]. Together, this drives the preferential trafficking of gut-primed T cells to intestinal tissues.

Accordingly, numerous efforts have been undertaken to interfere with immune cell trafficking in the intestine in order to develop therapeutics for IBD[11]. One of these approaches is the anti-β7 integrin antibody etrolizumab, which targets the β7 integrin subunit of the integrin heterodimers α4β7 and αEβ7. The latter mechanism is thought to inhibit the E-Cadherin-dependent mucosal retention of immune cells. Consistently, preclinical models demonstrated increased reduction of intestinal T cell accumulation upon β7 compared with α4β7 blockade[12,13].

[1]Department of Medicine 1, University Hospital Erlangen and Friedrich-Alexander-Universität Erlangen-Nürnberg, Erlangen, Germany. [2]Deutsches Zentrum Immuntherapie, University Hospital Erlangen, Erlangen, Germany. ✉e-mail: sebastian.zundler@uk-erlangen.de

However, a recent huge phase III trial program in UC did not convincingly demonstrate efficacy of etrolizumab for the induction and maintenance of remission[14]. Furthermore, in the phase III BERGAMOT trial in CD, etrolizumab was only effective during maintenance, but not during induction[15]. The question, why this is the case despite promising preclinical and phase II data[16], is still unresolved.

Previous studies had shown that αEβ7 expression in the ileum is higher than in the colon[17] and that α4β1-dependent pathways potentially bypass the blockade of α4β7-dependent gut homing in ileal CD[18]. It had therefore been suggested that targeting αEβ7 might be helpful to tackle cells that continue to home to the gut in the presence of vedolizumab at the tissue level, since this might compensate for redundant pathways overriding sole α4β7 blockade particularly in CD. However, this rationale to block αEβ7 in addition to α4β7 integrin has never formally addressed. Moreover, it has been shown in cancer and gastritis that the interaction of E-Cadherin with αEβ7 integrin also leads to the co-stimulation of T cells[19–21]. Yet, this aspect has not been investigated in the context of IBD so far and it is therefore unclear whether this mechanism contributes to the clinical effects observed with etrolizumab[22].

This work aims to further understand the mechanisms of etrolizumab in CD in the light of recent clinical observations. We explore pathways and mechanisms bypassing α4β7 blockade in T cells predestined to express αEβ7 and elucidate the role of E-Cadherin for the co-stimulation of these evading cells and the potential of etrolizumab to block it. While our data show that αEβ7 integrin is a reasonable target, etrolizumab fails to block co-stimulatory αEβ7-E-Cadherin interactions at high levels of T cell stimulation. Moreover, in patients with CD treated with etrolizumab, but not with placebo, cytotoxic activity in week 14 correlated to T cell activation.

Collectively, our data show that αEβ7 has functions beyond cell trafficking that have not been addressed in IBD so far and seem to remain intact in the presence of etrolizumab. Thus, our findings might help to explain observations from clinical trials and might guide future efforts to exploit β7 as a therapeutic target in IBD.

## Results

### Triple-integrin-expressing T cells are reduced in the peripheral blood of patients with CD

To understand whether blocking αEβ7 in addition to α4β7 is a reasonable therapeutic approach, we investigated the cell trafficking pathways that might guide cells to evade α4β7 blockade. Thus, we determined the expression of α4, β7 and β1 integrins on the surface of CD4$^+$ and CD8$^+$ memory T cells (Supplementary Fig. 1A) in the peripheral blood of non-IBD controls and patients with CD. We observed that around one half of the α4$^+$β7$^+$ cells co-expressed β1 (Supplementary Fig. 1B) and found a clear reduction of β7-expressing subsets in patients with CD (Fig. 1A, B). These data suggested that gut-homing T cells are depleted from the circulation probably due to recruitment to the intestine in CD and that a substantial portion of T cells expressing α4β7 might be able to alternatively employ α4β1 for gut homing in the presence of anti-α4β7 antibodies.

Accordingly, to study this possibility on a functional level, we performed dynamic adhesion assays with β7$^+$ memory T cells purified from the peripheral blood by fluorescence-activated cell sorting (FACS, Supplementary Fig. 2A). These cells were perfused through capillaries coated with MAdCAM-1 and vascular cell adhesion molecule (VCAM)−1 to mimic co-expression on the intestinal endothelium. Adhesion to the capillaries was only marginally reduced by vedolizumab both for CD4$^+$ and CD8$^+$ subsets. However, treatment with the pan-α4 integrin antibody natalizumab led to a profound reduction of adhesion indicating that the difference is due to the adhesion of T cells co-expressing α4β1 to VCAM-1 (Supplementary Fig. 2B, C).

In a separate approach, we also tested vedolizumab evasion during transmigration of β7-expressing CD4$^+$ or CD8$^+$ memory T cells over porous membranes coated with MAdCAM-1 and VCAM-1. Again, vedolizumab had small, but significant effects, whereas natalizumab very clearly reduced transmigration of CD4$^+$ as well as CD8$^+$ T cells (Supplementary Fig. 3).

Together with the expression data presented above, these findings supported the concept that a substantial subset of β7$^+$ memory T cells co-expresses α4β1, which is still functional for gut homing even in the presence of the anti-α4β7 integrin antibody vedolizumab, and that it might therefore be reasonable to additionally target these cells at the tissue level.

### Triple-integrin-expressing T cells decrease during vedolizumab treatment in patients that fail to achieve remission

To explore whether this might also be relevant in vivo, we examined the abundance of peripheral blood α4$^+$β7$^+$β1$^{hi}$ and α4$^+$β7$^+$β1$^{lo}$ CD3$^+$ T cells in patients with IBD that were treated with vedolizumab at baseline and prior to the fifth application (mean: 17 weeks). Intriguingly, we observed a selective reduction of α4$^+$β7$^+$β1$^{hi}$ T cells consistent with increased gut homing in patients that did not enter remission, while the dynamics of α4$^+$β7$^+$β1$^{lo}$ T cells did not differ between patients achieving remission or not (Fig. 2). Thus, alternative homing of triple-integrin-expressing T cells might be a mechanism contributing to failure of α4β7 blockade.

### Triple-integrin-expressing T cells have a marked cytotoxic phenotype

To estimate, which potential consequences the evasion of α4β7 blockade by these cells might have in the gut, we further characterized triple-integrin-expressing T cells. To this end, we sorted α4-expressing memory T cells from the peripheral blood of a control donor and patient with CD (Supplementary Fig. 4A, B) and submitted these cells to single cell RNA-sequencing (scRNA-seq). Unsupervised Leiden clustering of the pooled cells from both donors categorized the cells into six clusters (Fig. 3A). Based on visual representation, overall, cells from the patient with CD and the control donor seemed to be quite evenly distributed (Fig. 3B). Interestingly, cells co-expressing *ITGB1* and *ITGB7* (i.e., triple-integrin-expressing cells, since cells had been selected based on α4 expression) largely clustered separately from cells only expressing *ITGB7* throughout the dataset (Fig. 3C). In a next step, we systematically explored the distribution of these subsets within the different clusters. Indeed, we found an enrichment of *ITGB7*$^+$*ITGB1*$^+$ cells in cluster 2, whereas *ITGB7*$^+$*ITGB1*$^-$ cells were enriched in cluster 6 (Fig. 3D). These data suggested that cluster 2 might be particularly representative for triple-integrin-expressing T cells and we therefore calculated differential gene expression between cluster 2 and the pooled other clusters. Intriguingly, among the top 15 genes upregulated in cluster 2, seven genes including *GZMB*, *PRF1* and *GNLY* were found with direct cytotoxic function (Fig. 3E). Consistently, cells expressing these genes were predominantly located in cluster 2 (Fig. 3F and Supplementary Fig. 4C), but this did not result from an enrichment of CD8- over CD4-expressing T cells (Supplementary Fig. 4D). This promoted the notion that triple-integrin-expressing memory T cells are a subset with a marked cytotoxic signature. Indeed, when we directly compared the expression of genes associated with cytotoxicity between *ITGB7*$^+$*ITGB1*$^+$ and *ITGB7*$^+$*ITGB1*$^-$ cells, these genes were much more abundantly expressed in the former (Fig. 3G).

To validate this on protein level, we performed intracellular flow cytometry for Granzyme B and Granulysin with peripheral blood memory T cells from a cohort of patients with CD as well as healthy donors. Consistent with the transcriptomic data, α4$^+$β7$^+$β1$^{hi}$ cells expressed higher levels of these mediators than α4$^+$β7$^+$β1$^{lo}$ cells and the difference was more pronounced in CD than in controls (Fig. 3H).

Together, these findings indicated that the gut homing of triple-integrin-expressing cells with the potential to evade blockade by

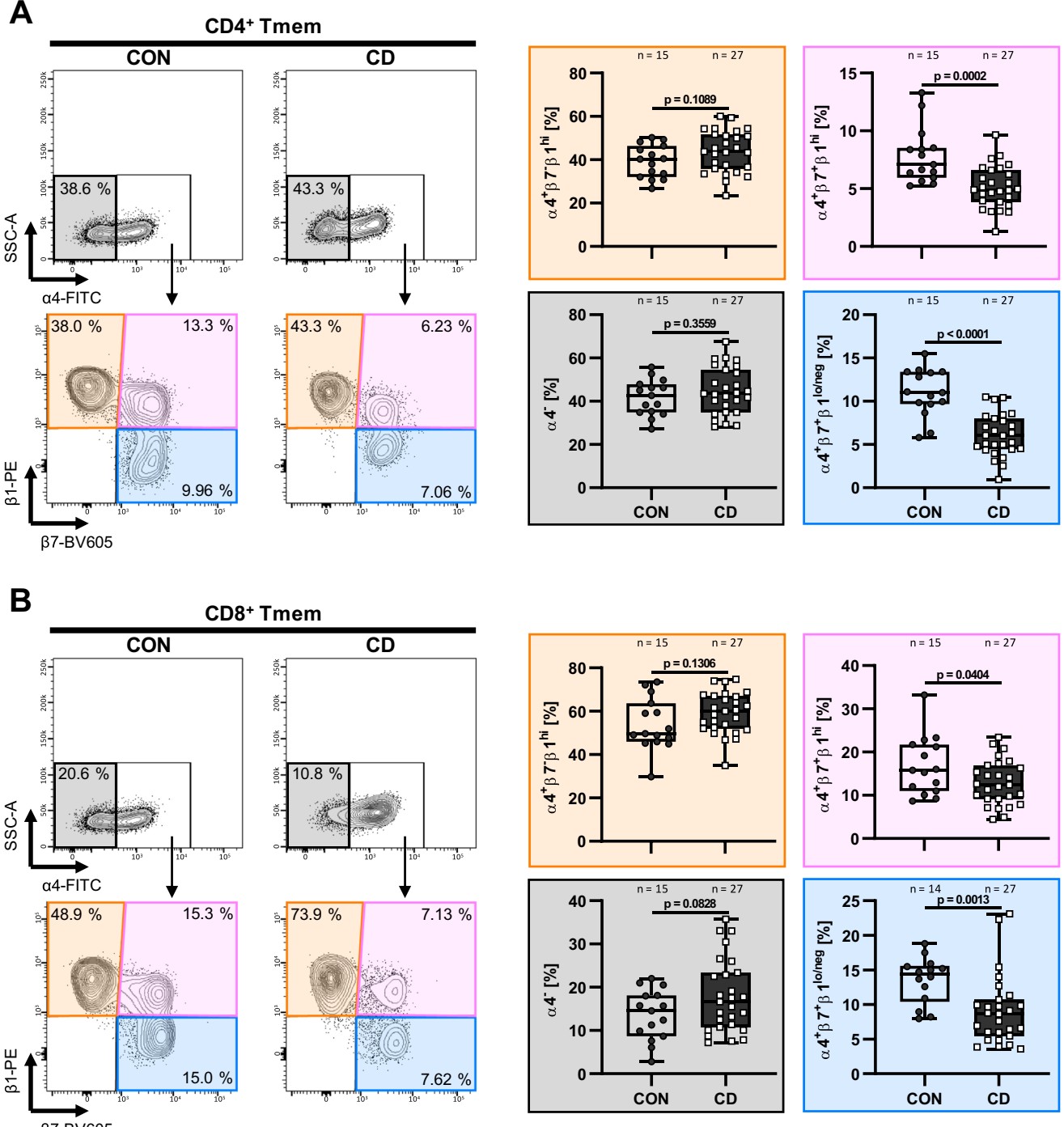

**Fig. 1 | Reduced proportions of β7⁺ peripheral memory T cells in Crohn's disease.** Representative (left) and quantitative (right) flow cytometry of α4, β7 and β1 expression on peripheral blood CD4⁺ (**A**) and CD8⁺ (**B**) memory T cells (Tmem) (CD3⁺TCRVα7.2⁻CD4⁺CD8a⁻CD45RA⁻ and CD3⁺TCRVα7.2⁻CD4⁻CD8a⁺CD45RA^lo/neg, respectively). $n = 14–27$ patients with Crohn's disease (CD) or non-IBD controls (CON). Significant outliers were detected by 1% ROUT test and removed from analysis. Normality was determined by D'Agostino and Pearson test and two-tailed unpaired $t$ or Mann–Whitney tests were performed accordingly. Data are displayed as box-whisker plots from minimum to maximum. The center line indicates the median with box limits defining the quartiles. Source data are provided as a Source Data file.

vedolizumab might drive cytotoxicity and inflammation in intestinal tissues.

**TGF-β induces αE integrin on a pro-inflammatory subset of β7⁺ memory T cells**

Since it is known that many T cells upregulate αE integrin (CD103) in response to TGF-β once homed to the gut[12,23] and thereby get a nominal target of etrolizumab, we wondered whether β7⁺ memory T cells evading vedolizumab blockade also have the potential to induce CD103.

To address this question, we incubated β7⁺ and β7⁻ CD4⁺ and CD8⁺ memory T cells from the peripheral blood with TGF-β and analyzed the effect on CD103 expression. TGF-β substantially increased CD103 expression in all four T cell subsets, but higher levels were found on β7⁺ compared to β7⁻ T cells and TGF-β-exposed CD8⁺β7⁺ memory T cells

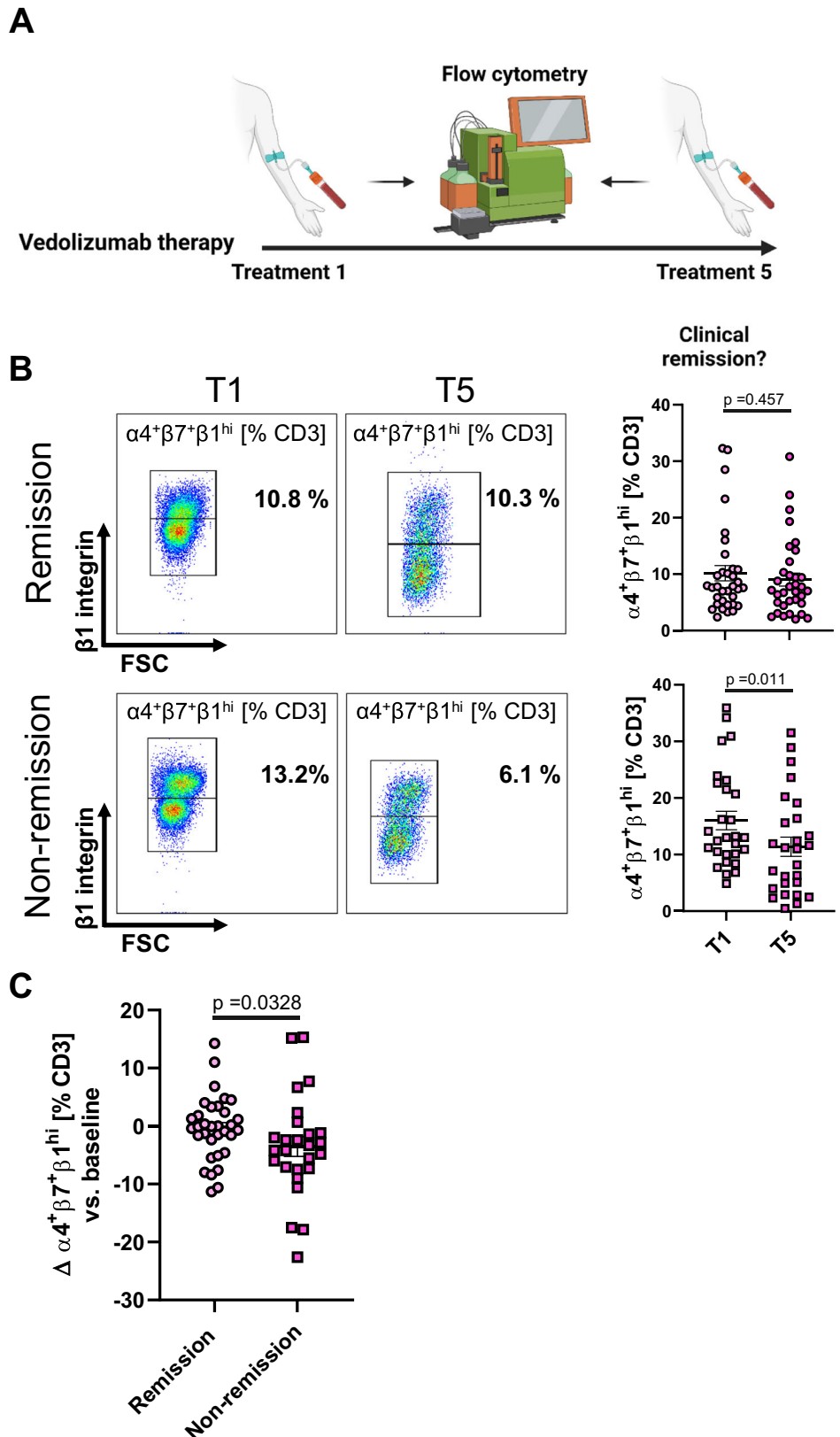

**Fig. 2 | Selective reduction of α4⁺β7⁺β1^hi T cells in patients with IBD that do not achieve remission with vedolizumab. A** Schematic depiction of the study. Peripheral blood from patients that underwent therapy with vedolizumab was analyzed by flow cytometry at baseline and at treatment five, when clinical remission was determined (HBI ≤ 4, PMS ≤ 1). Prepared with licensed BioRender application. **B** Representative and quantitative flow cytometry of the abundance of α4⁺β7⁺β1^hi T cells at baseline and at treatment five in patients with ($n = 34$) and without ($n = 27$) remission. **C** Difference in the abundance of α4⁺β7⁺β1^hi T cells at baseline and at treatment five in patients with remission or non-remission. Data are displayed as scatter plots with center lines and error bars depicting mean ± SEM. Source data are provided as a Source Data file. Two-tailed Wilcoxon (**B**) or Mann–Whitney tests (**C**) were performed.

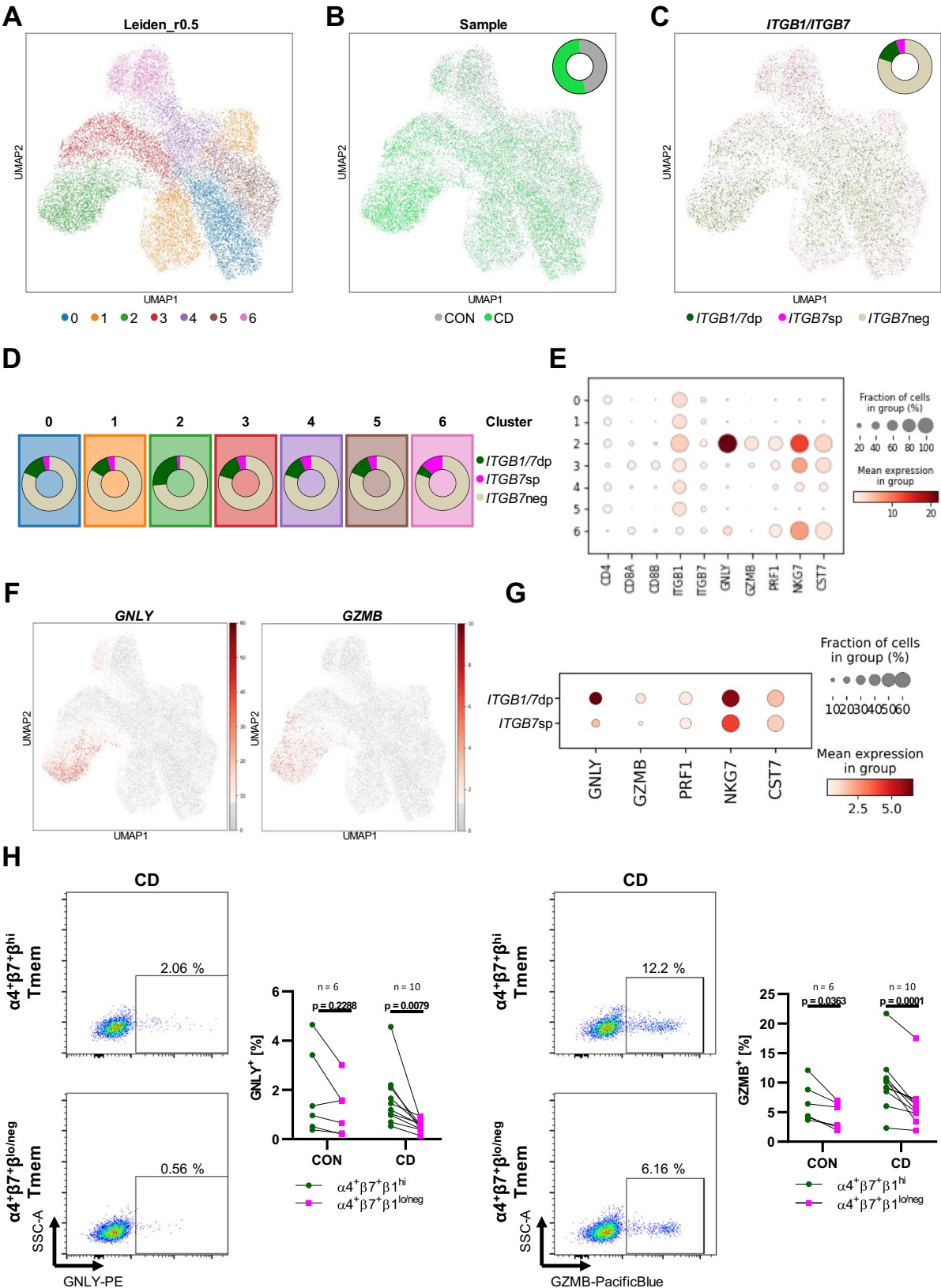

exhibited the highest levels (Fig. 4A–D). In a next step, we wanted to find out whether the potential of TGF-β to induce CD103 is preserved in β7-expressing memory T cells evading vedolizumab. Thus, we performed transmigration assays over porous membranes co-coated with MAdCAM-1 and VCAM-1, treated the cells with or without vedolizumab and tested the capacity of transmigrated cells to induce CD103 upon

treatment with TGF-β. Indeed, TGF-β led to a substantial increase of CD103 expression that was similar between cells that did or did not evade vedolizumab blockade (Fig. 4E–H). Thus, a subset of vedolizumab-evading T cells readily expresses αEβ7 integrin predisposing it to being targeted by anti-β7 antibodies such as etrolizumab in the tissue.

**Fig. 3 | Single cell RNA-sequencing identifies a cytotoxic signature of triple-integrin-expressing peripheral blood T cells in CD. A** UMAP plot showing unsupervised clustering of sorted α4⁺ memory T cells (Tmem) from the peripheral blood of a donor with CD and a non-IBD control (CON) based on Leiden algorithm at resolution 0.5. **B** UMAP plot showing the distribution of cells from CON (gray) and CD (light green). Pie chart illustrates the overall proportions. **C** UMAP plot showing the distribution of cell that do not express *ITGB7* (*ITGB7*neg, tawny), express *ITGB7*, but not *ITGB1* (*ITGB7*sp, magenta) or express *ITGB7* and *ITGB1* (*ITGB1/7*dp, dark green). **D** Pie charts illustrating the proportions of *ITGB7*neg, *ITGB7*sp, *ITGB1/7*dp cells per Leiden cluster. **E** Heatmap showing the fraction of cells expressing and the mean expression of the indicated genes. **F** UMAP plots showing the distribution of cells expressing *GNLY* (left) and *GZMB* (right). **G** Heatmap showing the fraction of cells expressing and the mean expression of the indicated genes in *ITGB1/7*dp and *ITGB7*sp cells. **H** Representative and quantitative flow cytometry of granulysin (GNLY, left) and granzyme B (GZMB, right) protein expression on peripheral blood CD3⁺CD45RA⁻α4⁺β7⁺ memory T cells, stratified according to their β1 expression, from CON and CD (*n* = 6–10). Significant outliers were detected by 1% ROUT test and removed from analysis. Two-way ANOVAs with Sidak's multiple comparisons test were performed. Source data are provided as a Source Data file.

Subsequently, we aimed to further corroborate the inflammatory potential of such CD103⁺ T cells. Accordingly, we induced CD103 expression on CD4⁺ and CD8⁺ T cells from healthy donors and patients with CD, rested them and determined cytokine expression by intracellular flow cytometry following re-stimulation. While CD103 expression was similar on the cells from healthy donors and CD (Supplementary Fig. 5A, B), the expression of IFN-γ, TNF-α and IL-17A, but not IL-2, were significantly increased on CD4⁺CD103⁺ T cells from patients with CD compared to controls (Fig. 5 and Supplementary Fig. 5D, F). In CD8⁺ T cells, we observed a significant increase in CD103⁺ cells co-expressing IFN-γ and IL-17A (Supplementary Fig. 5C, E). Moreover, the expression of all three cytokines was higher on CD4⁺CD103⁺ compared with CD4⁺CD103⁻ T cells in CD (Fig. 5A, B) and more CD8⁺CD103⁺ than CD8⁺CD103⁻ T cells from patients with CD expressed IL-17A or co-expressed IL-17A and IFN-γ (Supplementary Fig. 5E). Together, and in consistence with the literature[23], these data indicated that human CD103⁺ T cells are a particularly pro-inflammatory subset. In conclusion, β7⁺ memory T cells evading gut homing blockade by vedolizumab via VCAM-1 might develop into a pro-inflammatory CD103⁺ T cell subset upon exposure to TGF-β in the gut. This substantiated the notion that, in principle, it is reasonable and feasible to target these cells on mucosal level with anti-β7 antibodies.

## Etrolizumab-s blocks E-Cadherin-dependent T cell co-stimulation at low, but not at high levels of T cell stimulation

Based on this rationale, we interrogated whether etrolizumab is useful for such targeting. Since the impact of etrolizumab for T cell retention has previously been characterized, we particularly focused on the question whether αEβ7-E-Cadherin interactions drive T cell co-stimulation and how etrolizumab might interfere with this process.

To this end, we developed a co-stimulation assay using plate-coated E-Cadherin together with CD8⁺ T cells previously exposed to TGF-β to induce CD103 expression (Fig. 6A). T cell receptor stimulation was accomplished by an anti-CD3 antibody. Indeed, the presence of E-Cadherin substantially increased the expression of CD69 at two different concentrations of the anti-CD3 antibody (Fig. 6B) indicating a co-stimulatory role of E-Cadherin for T cells. Treatment with the etrolizumab surrogate (-s) antibody FIB504 (sharing the antigen recognition site with etrolizumab) repressed upregulation of CD69 at low anti-CD3 concentrations confirming that the effect is E-Cadherin-αEβ7-dependent (Fig. 6B, C). However, the effect of etrolizumab-s was lost at high concentrations of anti-CD3 (Fig. 6B, D) suggesting that targeting β7 with etrolizumab is not sufficient to block E-Cadherin-mediated T cell co-stimulation, when T cells are highly stimulated. This notion was further corroborated by specific analysis of CD103-expressing T cells in the co-stimulation assays, which also showed a significantly reduced effect of etrolizumab-s at high anti-CD3 concentrations (Supplementary Fig. 6A–C).

Importantly, when we repeated our assays with a neutralizing anti-CD103 antibody, we observed a substantial and significant inhibition of E-Cadherin-dependent T cell co-stimulation regardless of the anti-CD3 concentration used (Supplementary Fig. 6D–F). Interestingly, and in consistence with previous literature[24], we observed that etrolizumab-s

only led to a gradual downregulation of surface CD103 levels (Supplementary Fig. 6G). Together, this strongly indicated that the limited potential of etrolizumab-s to inhibit T cell co-stimulation via αEβ7 and E-Cadherin is antibody-specific.

Next, we sought to determine, what this implies on a functional level. To this end, we performed co-stimulation assays as described above and quantified the expression of cytotoxic mediators in the supernatant. Intriguingly, the presence of E-Cadherin during T cell activation substantially increased the secretion of Perforin, Granzyme B, soluble Fas ligand and Granulysin. In analogy to our above data, etrolizumab-s abrogated this effect at low, but not at high anti-CD3 concentrations (Fig. 6E). In additional assays, we used etrolizumab-s at a clearly supraphysiological concentration together with high anti-CD3 concentrations to see whether this is able to restore the blockade of T cell co-stimulation. Indeed, this led to a modest, but significant reduction of the secretion of cytotoxic mediators (Supplementary Fig. 7) suggesting that very high etrolizumab-s exposure might partly compensate the resistance of T cells to interference of etrolizumab-s with αEβ7-dependent co-stimulation.

Collectively, these findings suggested that the interaction of αEβ7 integrin with E-Cadherin has important co-stimulatory functions in the context of CD, while etrolizumab fails to mitigate such activation in the presence of high levels of T cell stimulation as they occur in active inflammation.

## High T cell activation in patients treated with etrolizumab but not with placebo is associated with high expression of cytotoxic mediators at week 14

To explore whether these findings also reflect in human patients treated with etrolizumab, we analyzed bulk RNA sequencing data from ileal samples of the cohort 1 of the BERGAMOT phase III trial of etrolizumab in CD[13]. Based on week 0 and 14 expression of five genes associated with T cell activation, we calculated a T cell activation score in the 34 patients treated with etrolizumab and the 8 patients treated with placebo, which were available for analysis. Patients with a score of 6 or higher were considered to have high T cell activation, those with a score of 4 or lower to have low T cell activation (Fig. 7A, B). Consistently, the expression of T cell cytokines such as *IFNG*, *IL17F* or *IL22*, but not of innate cytokines such as *IL12A*, *IL18* or *IL15*, was increased in the "high" compared with the "low" group (Fig. 7C).

We then compared the expression of genes of cytotoxic mediators at week 14 between patients treated with etrolizumab with high and low T cell activation scores and found a higher expression in the former group (Fig. 7D). Importantly, *GNLY*, *GZMB* and *GZMH* were specifically decreased in etrolizumab-treated patients with low T cell activation, but not in placebo-treated patients and the absolute levels in etrolizumab-treated patients with high T cell activation were similar to that of placebo-treated patients (Fig. 7E). Moreover, *FASLG* was specifically downregulated from week 0 to 14 in etrolizumab-treated patients with low T cell activation, but not in those with high T cell activation or treated with placebo (Supplementary Fig. 8). Thus, these data were consistent with the notion that, in line with our in vitro data, etrolizumab is able to reduce cytotoxic mediators in an environment with low, but not with high T cell stimulation.

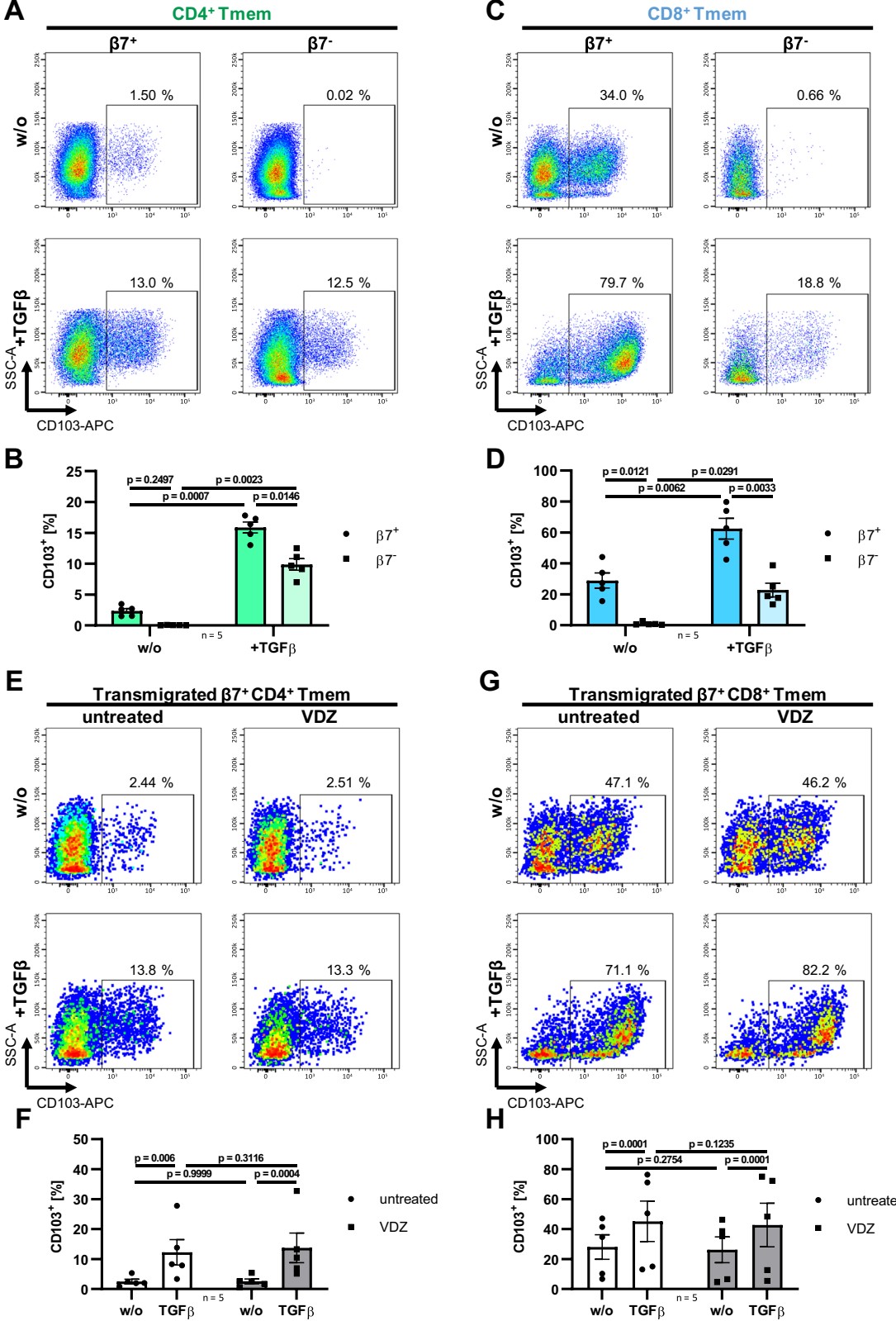

## Discussion

The results of the phase III trial program of etrolizumab in UC and CD were overall disappointing. While the primary endpoint of clinical remission at week 10 or 14 was achieved in two out of three induction trials in UC, none of the primary endpoints was met in maintenance trials. However, several key secondary endpoints were met and efficacy was similar to anti-TNF antibodies in head-to-head comparisons[15,25–28].

In Crohn's disease, efficacy was only observed in maintenance, but not during induction.

A compelling explanation for these observations is missing so far, since they are in contrast to convincing phase II data in ulcerative colitis and findings in preclinical models suggesting superior reduction of intestinal T cell accumulation by anti-β7 compared to anti-α4β7 treatment[12,13]. Although it has been suggested that αEβ7 might not be a

**Fig. 4 | β7⁺ memory T cells (Tmem) that evade blockade by vedolizumab (VDZ) readily induce CD103.** Representative and quantitative flow cytometry of CD103 expression on sorted β7⁺ and β7⁻ CD4⁺ Tmem (CD3⁺TCRVα7.2⁻CD4⁺CD8a⁻CD45RA⁻β7⁺/⁻) (**A**, **B**) and on sorted β7⁺ and β7⁻ CD8⁺ Tmem (CD3⁺TCRVα7.2⁻CD4⁻CD8a⁺CD45RA⁻β7⁺/⁻) (**C**, **D**) after 3 days of incubation with anti-CD2/3/28 stimulation beads in the presence or absence (w/o) of 10 ng/ml TGFβ1. $n = 5$.
**E–H** Representative and quantitative flow cytometry of CD103 expression on sorted β7⁺ CD4⁺ Tmem (CD3⁺TCRVα7.2⁻CD4⁺CD8a⁻CD45RA⁻β7⁺) (**E**, **F**) and on sorted β7⁺ CD8⁺ Tmem (CD3⁺TCRVα7.2⁻CD4⁻CD8a⁺CD45RA⁻β7⁺) (**G**, **H**) treated with or without (untreated) 10 μg/ml VDZ before transmigration over a MAdCAM-1- and VCAM-1-coated membrane and subsequent incubation with anti-CD2/3/28 stimulation beads in the presence or absence of 10 ng/ml TGFβ1. $n = 5$. Significant outliers were detected by 1% ROUT test and removed from analysis. Two-way ANOVAs with Tukey's multiple comparisons test were performed. Data are displayed as bar plots depicting mean ± SEM. Source data are provided as a Source Data file.

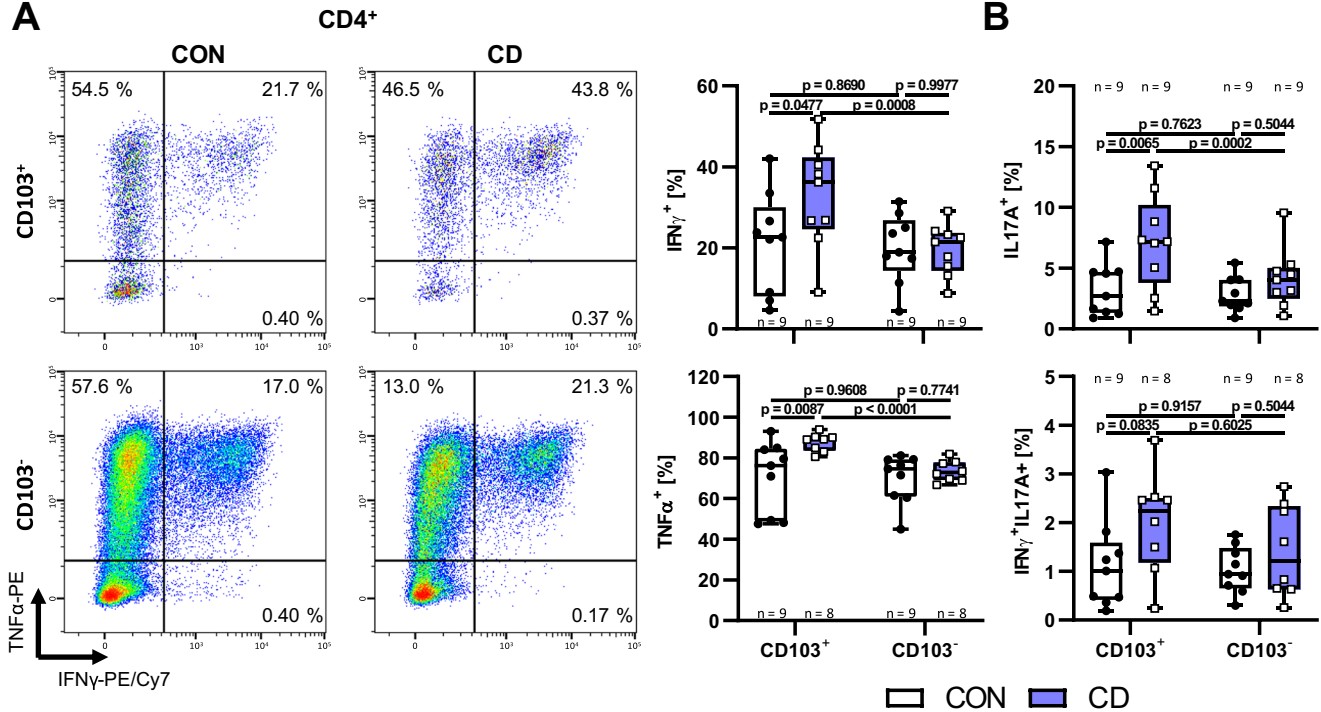

**Fig. 5 | CD103-expressing T cells are proficient producers of pro-inflammatory cytokines.** Representative and quantitative flow cytometry of IFNγ and TNFα expression (**A**) and quantitative flow cytometry of IL17A as well as IL17A and IFNγ co-expression (**B**) by stimulated and TGFβ1-treated CD103⁺/⁻ peripheral blood CD4⁺ T cells. $n = 7$–9 patients with Crohn's disease (CD) or non-IBD controls (CON) as indicated. Significant outliers were detected by 1% ROUT test and removed from analysis. Two-way ANOVAs with Sidak's multiple comparisons test were performed. Data are displayed as box-whisker plots from minimum to maximum. The center line indicates the median with box limits defining the quartiles. Source data are provided as a Source Data file.

suitable target[29], since some studies had found decreased expression in active IBD[30,31], the current and previous studies[13,23] have shown that human CD103⁺ T cells have a pro-inflammatory phenotype. Thus, regardless of the numbers present in the gut, one would assume that counteracting a pro-inflammatory T cell subset should be beneficial in active inflammation.

When considering potential approaches to explain the clinical trial results from a mechanistic perspective, it is necessary to separately understand the action on α4β7 and αEβ7. With regard to α4β7, it is beyond doubt that etrolizumab efficiently blocks gut homing, since more β7-expressing memory T cells can be found in the peripheral circulation under treatment[16] and functional in vitro assays have directly shown that etrolizumab abrogates α4β7-MAdCAM-1 interaction[32]. However, there are clues that the effect of etrolizumab on T cell subsets is different from vedolizumab, which preferentially targets effector T cells in a certain exposure range[33,34].

With regard to αEβ7, post hoc analyses of the phase II trial in UC had suggested that this aspect is particularly relevant for the in vivo action of etrolizumab, since high baseline expression levels of αE predicted response to treatment[35]. Moreover, activity of etrolizumab against epithelial retention via E-Cadherin can be assumed based on in vitro and experimental in vivo findings[12,13]. The data presented in this study now extend the focus to previously unrecognized aspects of the impact of etrolizumab on αEβ7-E-Cadherin interaction. Our findings demonstrate that E-Cadherin co-stimulates CD8⁺ T cells as evident by the upregulation of CD69 and the secretion of cytotoxic mediators and suggest that this mechanism might be important in the context of IBD. Together with a previous report showing that the interaction of αEβ7 with E-Cadherin also drives T cell infiltration to epithelial organoids[36], it emerges that αEβ7 is a key modulator of epithelial-lymphocyte cross-talk. However, while etrolizumab seems to be able to block adhesive lympho-epithelial interaction and cytotoxicity resulting from direct cell-to-cell contact of T lymphocytes with the epithelium[36], our data suggest that co-stimulatory pathways leading to the secretion of soluble cytotoxic mediators are not sufficiently blocked and we therefore hypothesize that these mediators might cause epithelial cell death in the inflamed gut in a paracrine manner as demonstrated in the literature[37,38].

Our data on the co-stimulatory role of αE are well in line with previous findings for CD8⁺ T cells in the field of cancer[20,21] and CD4⁺ T cells in *Helicobacter pylori* gastritis[19]. Interestingly, while etrolizumab-s was able to abrogate this co-stimulation in an environment with low T cell receptor stimulation, it failed to do so in the context of high T cell receptor stimulation. Previous data have shown that while etrolizumab-s induces the internalization of α4β7 integrin

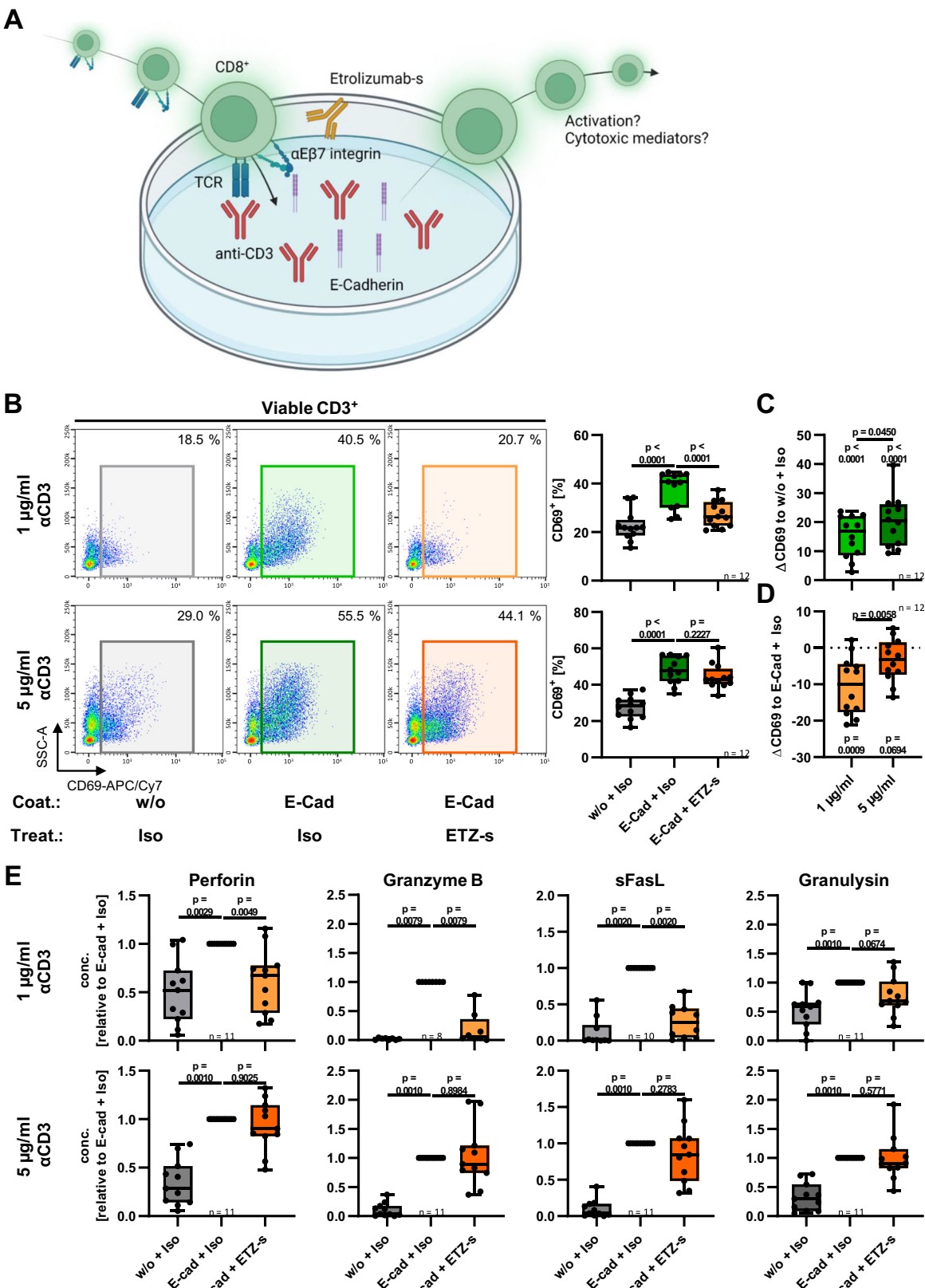

from the T cell surface even more efficiently than vedolizumab, αEβ7 integrin is only partly internalized[24]. Therefore, the remaining molecules might be sufficient to drive co-stimulation in the presence of high T cell receptor stimulation.

This was also suggested by an analysis of RNA sequencing data from patients with CD treated with etrolizumab in a cohort of the BERGAMOT phase III trial. It must be acknowledged that this analysis has several limitations, especially due to difficulties in drawing conclusions on specific immune cell populations based on bulk RNA sequencing. However, since etrolizumab is not available outside clinical trials, it is currently not possible to obtain direct evidence, e.g., by characterizing immune cells from biopsies. Thus, the lack of analyses

**Fig. 6 | Limited efficacy of etrolizumab surrogate (ETZ-s) to block E-Cadherin-dependent co-stimulation of CD8[+] T cells. A** Schematic depiction of the experimental setup in the co-stimulation assays. Magnetically enriched peripheral blood CD8[+] T cells were stimulated for 3 days with anti-CD2/3/28 beads in the presence of 10 ng/ml TGFβ1, rested for 1 day, and subsequently re-stimulated on plates coated with either 1 or 5 μg/ml anti-CD3 together with or without (w/o) E-Cadherin (E-Cad) in the presence of 10 μg/ml ETZ-s or an isotype control (Iso). Prepared with licensed BioRender application. **B** Representative and quantitative flow cytometry of CD69 expression after re-stimulation. Significant outliers were detected by 1% ROUT test and removed from analysis. Normality was determined by D'Agostino and Pearson test and repeated measures one-way ANOVAs with Dunnett's multiple comparisons test were performed accordingly. Change of CD69 expression compared to w/o + Iso (**C**) and change of CD69 expression compared to E-Cad + Iso (**D**). Significant outliers were detected by 1% ROUT test and removed from analysis. Normality was determined by D'Agostino and Pearson test and two-tailed paired and one sample *t* tests were performed accordingly. *n* = 12. **E** Relative concentrations of the indicated proteins in the supernatants of re-stimulated cultures relative to E-Cad + Iso. Significant outliers were detected by 1% ROUT test and removed from analysis. Two-sided one sample Wilcoxon signed rank tests were performed. *n* = 8–11. Data are displayed as box-whisker plots from minimum to maximum. The center line indicates the median with box limits defining the quartiles. Source data are provided as a Source Data file. TCR T cell receptor.

on tissue biopsies from patients treated with etrolizumab needs to be considered as an—unfortunately uncurable—caveat, when interpreting our data. Moreover, since data on response or non-response of these patients to etrolizumab were not available, we were not able to correlate our findings to clinical outcomes, which would have provided important additional validation.

Yet, several clinical observations are interesting in the context of our data and support the relevance of our findings. A recent post hoc analysis of covariates associated with response to etrolizumab in the various clinical trials revealed that low baseline Mayo Clinic Score was linked to induction of remission and endoscopic improvement[39]. While clinical disease activity is of course influenced by many factors, the activation of T cells, which play a central role in the pathogenesis of IBD[40], might be one of them. Thus, these patients may have been predisposed for an impact of etrolizumab on E-Cadherin-dependent co-stimulation and associated benefit for their clinical situation. The efficacy observed with etrolizumab therapy during maintenance treatment in CD[15] might point into a similar direction, since these are patients that had a response during induction and, thus, started maintenance therapy with low inflammation levels. However, it must be noted that improved efficacy with lower baseline disease activity is not an exclusive feature of etrolizumab, but has also been reported for vedolizumab[41,42].

Taken together, the points mentioned are indeed compatible with suboptimal clinical efficacy of etrolizumab and our data represent an important approach to explain the clinical observations in the phase III trial program of etrolizumab. As such, they also underscore, that it is important to consider pleiotropic effects of candidate targets and antibody-specific potency to block these effects.

At the same time, our data also substantiate β7 and/or αE integrin as plausible targets in IBD, since targeting αEβ7 at the tissue level might help to hamper the activation of pro-inflammatory cells that fall through the cracks of α4β7 blockade due to co-expression of functionally active redundant gut-homing molecules. Our data point to the possibility that it might be even more promising to block αE to this end, since the ability of an anti-αE antibody to block co-stimulation was preserved at high levels of T cell stimulation. However, it is also perceivable that the same effect can be achieved with anti-β7 antibodies binding to different epitopes than etrolizumab. Thus, further translational research must clarify whether other anti-β7 antibodies or anti-αE antibodies might be worth another attempt to exploit αEβ7 integrin-dependent mechanisms for the treatment of IBD.

## Methods

### Patients, blood samples and outcomes
For flow cytometry experiments, fluorescence-activated cell sorting (FACS) of α4[+] memory T cells and experiments requiring magnetic subset enrichment, EDTA-anticoagulated blood from non-IBD controls and patients with Crohn's disease was used. These samples were obtained from the IBD Outpatient Clinic of the Department of Medicine 1 of the University Hospital Erlangen, Germany. Donor characteristics are summarized in Supplementary Table 1. For FACS of β7[+] memory T cells, leukocyte cones were obtained from the Department of Transfusion Medicine and Haemostaseology of the University Hospital Erlangen,

Germany. For the longitudinal assessment of β1[hi] and β1[lo] subsets of α4[+]β7[+] T cells in patients treated with vedolizumab, blood was obtained from patients with active IBD (Harvey-Bradshaw-Index (HBI) ≥ 5 for CD; partial Mayo score (PMS) ≥ 2 for UC; and/or steroid dependency) at baseline and prior to the fifth application of vedolizumab (mean 17 weeks). Baseline characteristics are listed in Supplementary Table 2. Clinical remission at treatment five was defined as PMS ≤ 1 or HBI ≤ 4.

All samples were obtained following informed written consent and according to approval of the Ethics Committee of the Friedrich-Alexander-Universität Erlangen-Nürnberg.

### Isolation of peripheral blood mononuclear cells (PBMCs)
PBMCs were isolated with Pancoll-based (PAN-Biotech) density gradient centrifugation.

### Flow cytometry
The following fluorochrome-conjugated antibodies were used for extracellular staining using standard protocols, including Fc receptor blocking (Miltenyi Biotec): CD3 (BUV395, SK7, BD Biosciences (1:100)/APC, HIT3a, Biolegend (1:100)/FITC, OKT3, Biolegend (1:100)/VioGreen, REA613, Miltenyi Biotec/BV510, OKT3, Biolegend), CD4 (BUV496, SK3, BD Biosciences (1:100)/VioBlue, VIT4, Miltenyi Biotec), CD8a (PerCP/Cy5.5, RPA-T8, Biolegend (1:100)), CD25 (FITC, M-A251, Biolegend (1:100)), CD45RA (VioGreen, REA1047, Miltenyi Biotec (1:200)/APC/Cy7, HI100, Biolegend (1:100)), CD69 (APC/Cy7, FN50, Biolegend (1:100)), CD103 (APC/PE/Cy7, Ber-ACT8, Biolegend (1:100)), CCR9 (PE/Cy7, L053E8, Biolegend (1:100)), TCRVα7.2 (BV421, OF5A12, BD Biosciences (1:50)), α4 (FITC, MZ18-24A9, Miltenyi Biotec (1:125)/PE/Cy7, 9F10, Biolegend (1:200)/VioBlue, MZ18-24A9, Miltenyi Biotec/BV421, 9F10, Biolegend), β1 (AF488 (1:200)/PE (1:100), TS2/16, Biolegend), β7 (BV605, FIB504, BD Biosciences (1:33)/PE, FIB27, Biolegend (1:100)/APC, TS2/16, Biolegend). In some experimental setups, cells were fixed using BD CellFix (BD Biosciences) prior to acquisition.

When intracellular staining was required, the following fluorochrome-conjugated antibodies were used in combination with the Foxp3/Transcription Factor Staining Buffer Set (eBioscience) after extracellular staining: GNLY (PE, DH2, Biolegend (1:100)), GZMB (PacificBlue, GB11, Biolegend (1:100)), GZMK (APC, GM26E7, Biolegend (1:100)), IFNγ (PE/Cy7, B27, Biolegend (1:100)), IL2 (BV421, 5344.111, BD Biosciences (1:100)), IL17A (PE, BL168, Biolegend (1:100)), TNFα (PE, Mab11, Biolegend (1:100)).

In stimulation experiments, Fixable Viability Dye eFluor 506 or eFluor780 (eBioscience (1:1000)) was used for exclusion of dead cells.

LSRFortessa (BD Bioscience), MACSQuant 10 and MACSQuant 16 (Miltenyi Biotec) instruments were used for data acquisition. FlowJo v10.7.1 (Tree Star) was used for data analysis.

### Fluorescence- and magnetically-activated cell sorting (FACS/MACS)
FACS was performed to isolate α4[+] memory T cells for subsequent scRNASeq, with α4[+] memory T cells being defined as CD3[+]CD45RA[−]α4[+] viable single lymphocytes. To that end, the following fluorescently-labeled antibodies were used in combination with fixable viability dye

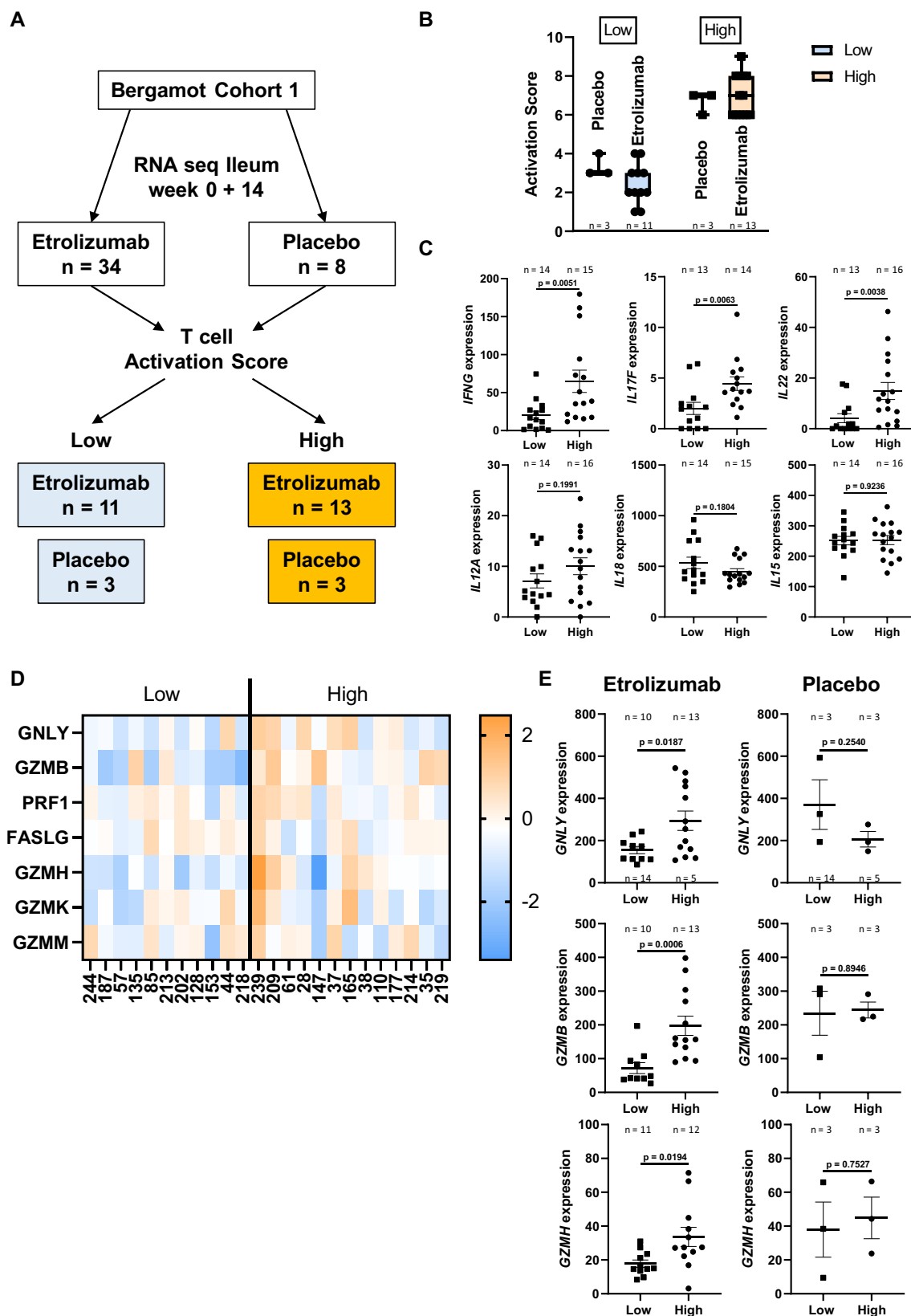

eFluor780 (eBioscience (1:1000)): CD3 (APC, HIT3a, Biolegend (1:100)), α4 (PE/Cy7, 9F10, Biolegend (1:100)), CD45RA (VioGreen, REA1047, Miltenyi Biotec (1:100)).

To isolate $\beta7^{+/-}$ CD4 ($CD3^+TCRV\alpha7.2^-CD4^+CD8a^-CD45RA^-$ $\beta7^{+/-}$ single lymphocytes) and CD8 ($CD3^+TCRV\alpha7.2^-CD4^-CD8a^+CD45RA^-$ $\beta7^{+/-}$ single lymphocytes) memory T cells for functional investigations

and CD103-inducibility analysis, we used fluorescently-labeled antibodies against the following epitopes: CD3 (PE/Cy7, SK7, Biolegend (1:1000)), TCRVα7.2 (BV421, OF5A12, BD Biosciences (1:500)), CD4 (APC-Vio770, VIT4, Miltenyi Biotec (1:2000)), CD8a (FITC, RPA-T8, Biolegend (1:2000)), CD45RA (VioGreen, REA1047, Miltenyi Biotec (1:1000)), β7 (PE, FIB27, Biolegend (1:1000)).

**Fig. 7 | Cytotoxic gene expression is selectively reduced in etrolizumab-treated patients with presumably low T cell activation. A** Schematic depiction of transcriptome analysis on patients treated with etrolizumab or placebo in Bergamot cohort 1 (GSE152316). Patients for which matched week 0 and 14 samples from the ileum were available were selected. Based on the expression of *CD40LG*, *TNFRSF4*, *IL2RA*, *CD69*, *TNFRSF9* a T cell activation score was calculated and patients stratified into presumed low or high T cell activation. **B** T cell activation scores in the indicated groups. Data are displayed as box-whisker plots from minimum to maximum. The center line indicates the median with box limits defining the quartiles. **C** Expression of the indicated genes in patients with low or high T cell activation scores. **D** Heat map of the expression of genes coding for cytotoxic mediators at week 14 in patients treated with etrolizumab with high or low activation scores. **E** Expression of the indicated genes of cytotoxic mediators at week 14 in patients with high or low activation scores and treated with etrolizumab or placebo. Significant outliers were detected by 1% ROUT test and removed from analysis. Normality was determined by D'Agostino and Pearson ($n \geq 10$) or Shapiro–Wilk ($n < 10$) test and two-tailed unpaired $t$ or Mann–Whitney tests were performed accordingly. Data are displayed as dot plots indicating mean ± SEM. Source data are provided as a Source Data file.

FACS Aria II SORP (BD Biosciences) instruments were used for FACS.

MACS enrichment for CD4 and CD8 T cells from PBMCs was performed according to manufacturer's instructions using the CD4 T Cell Isolation Kit, human and CD8 T Cell Isolation Kit, human, respectively (both Miltenyi Biotec).

### Dynamic adhesion assays to MAdCAM-1 and VCAM-1

For the quantification of dynamic adhesion in the presence of a combination of MAdCAM-1 and VCAM-1, an established protocol was adapted[34]. In short, FACS-purified $\beta 7^+$ CD4 and CD8 memory T cells were rested overnight in RPMI 1640 with 10% FCS and 1% P/S and subsequently incubated with Cell Trace CFSE (Invitrogen) for 15 min at 37 °C. Rectangle miniature capillaries (CM Scientific) were coated with a combination of 5 µg/ml of each MAdCAM-1- (R&D Systems) and VCAM-1-Fc Chimeras (Biolegend) in capillary coating buffer (150 mM NaCl, 1 mM 4-(2-hydroxyethyl)-1-piperazineethanesulfonic acid) for 1 h and then blocked with 10% FCS in phosphate buffered saline (PBS). Labeled cells were resuspended in adhesion buffer (150 mM NaCl, 1 mM CaCl₂, 1 mM MgCl₂) at a concentration of $0.75*10^6$/ml. Subsequently, 1 mM MnCl₂ was added to the cells and the cell solutions were perfused through the coated capillaries at 10 µl/min using a peristaltic pump (Shenchen). Afterwards, capillaries were rinsed at 50 µl/min to remove non-adherent cells and, finally, capillaries were imaged using an inverted microscope (Leica). Twelve images were acquired per capillary and adhering cells were analyzed using Fiji (National Institute of Health).

### Transwell transmigration assays

To analyze the ability of VDZ to inhibit $\alpha 4\beta 7$-dependent transmigration via MAdCAM-1 in the presence of VCAM-1, FACS-purified $\beta 7^+$ CD4 and CD8 memory T cells were resuspended in X-Vivo15 medium (Lonza) and preincubated with either 10 µg/ml VDZ or NTZ or left untreated for 20 min at 37 °C. After the addition of 1 mM MnCl₂, 75,000 cells were added to a 3 µm transwell insert (Corning), which was previously coated with either 5 µg/ml of MAdCAM-1, VCAM-1 or a combination of both for 1 h, in duplicates. Subsequently, the insert was placed into wells containing X-Vivo15 medium supplement with 10% FCS and incubated for 4 h at 37 °C. After incubation, inserts were discarded and transmigrated cells were enumerated by flow cytometry. To analyze the inducibility of CD103 on transmigrated cells, FACS-purified $\beta 7^+$ CD4 and CD8 memory T cells were starved overnight in X-Vivo15 medium. After starving, the assay was performed analogously, except for omitting MnCl₂ to retain viability. After the incubation, contents of 6–8 corresponding inserts and wells were pooled and transmigrated cells were labeled with Cell Trace CFSE for 15 min at 37 °C. Afterwards, labeled cells were reunited with the corresponding unstained non-transmigrated fraction. Cells were then resuspended in full RPMI medium, counted, and mixed with anti-CD2/3/28 activation beads (Miltenyi Biotec) in a 1:2 bead to cell ratio. After 3 days of culture at 37 °C in the presence or absence of 10 ng/ml rhTGFβ1 (Miltenyi Biotec), these cultures were analyzed via flow cytometry.

### In vitro stimulations

To analyze CD103-induced T cells, specified cell populations were first resuspended in full RPMI, mixed with anti-CD2/3/28 activation beads (Miltenyi biotec) in a 1:2 bead to cell ratio and cultured for 3 days in the presence or absence of 10 ng/ml rhTGFβ1 (Miltenyi Biotec) to induce CD103 expression.

To quantify cytokine expression capabilities, stimulation beads were removed from the cultures as per manufacturer's instructions and cells were rested for 24 h in full RPMI 1640 medium (+10% FCS, +1% P/S, Gibco). Afterwards, rested cells were restimulated for 4 h with 1 mM inomycin (Cayman Chemical) and 50 ng/ml PMA (Sigma) in the presence of a combination of Brefeldin A (Applichem) and Monensin (Biolegend) at a concentration of 0.5 µg/ml each, to inhibit cytokine release.

To elucidate CD103-dependent interactions with E-Cadherin, induced cells were rested in the same way. Subsequently, cells were restimulated in the presence of 10 µg/ml (or 100 µg/ml where specifically indicated) anti-β7 (FIB504, CellXGene), anti-CD103 (Ber-ACT8, Biolegend) or corresponding isotype controls (Sigma Aldrich, Biolegend) by transferring them to wells previously coated with anti-CD3 (UCHT2, Biolegend) at concentrations of 1 or 5 µg/ml with or without E-Cadherin Fc Chimeras (Biolegend) at a concentration of 5 µg/ml for 1 h. Plates were then cultured for 3 days at 37 °C.

### Single cell RNA sequencing

Single cell suspensions obtained from FACS as mentioned above were loaded in the 10x Chromium Controller and cDNA single-cell library was constructed using the dual index Chromium Single Cell 3' v3.1 protocol of 10x Genomics, Inc. The cDNA library was sequenced with GeneWiz LLC using an Illumina platform. Raw reads were passed to the 10x CellRanger software v7.0.0. The obtained UMI count matrix was analyzed with scanpy[43] v1.7.2 using Jupyter Notebook 1.0.0 on Python. All cells with a mitochondrial content higher than 20% were filtered out. The entire dataset was normalized with the SizeFactors function of DeSeq2 v1.24.0 in R v3.6.1. The normalized data was elog(x + 1) scaled with the log1p function of scanpy. The normalized and scaled data was reduced into 40 components using the Principal Component Analysis. These 40 components were further reduced into 2 dimensions using the Uniform Manifold Approximation and Projection (UMAP) algorithm[44]. The neighbor connectivities calculated by UMAP were passed to the Leiden algorithm[45] to detect communities with a resolution of 0.5.

### Bulk RNA sequencing

We re-analyzed previously published RNA sequencing data (GSE152316) from the cohort 1 of the BERGAMOT phase III trial in CD and specifically focused on paired samples from week 0 and week 14 obtained from the terminal ileum of patients treated with etrolizumab or placebo. Reads from these samples were normalized together using DeSeq2 v1.24.0 in R v3.6.1.

To estimate T cell activation in the samples we composed a score taking into account the transcripts of the five T cell activation-associated genes *CD40LG*, *TNFRSF4*, *IL2RA*, *CD69*, *TNFRSF9*. Expression of these genes at week 0 and week 14 was compared to the respective median expression in all samples of the cohort and patients received a score of 1 for every of the genes at both timepoints, which had a higher than median expression. Patients with a cumulative score of 6 or higher that did not score less than 2 on any time point were considered

as having high T cell activation, while patients with a cumulative score of 4 or lower that did not score more than 3 on any time point were considered as having low T cell activation.

## Statistics

Significant outliers were identified using ROUT test and excluded from the analysis. Data was tested for normality by D'Agostino and Pearson or Shapiro–Wilk test ($n < 10$). For statistical analyses, unpaired $t$ or Mann–Whitney tests, paired $t$ or Wilcoxon matched pairs signed rank tests, two-way ANOVA with Tukey's or Sidak's multiple comparison tests and one-way RM ANOVA with Dunnett's multiple comparison tests were calculated for absolute comparisons. For relative comparisons, one sample $t$ or Wilcoxon matched pairs signed rank tests were calculated.

## Reporting summary

Further information on research design is available in the Nature Portfolio Reporting Summary linked to this article.

## Data availability

Single cell transcriptomic data generated in this study have been deposited in the GEO database under the accession code GSE252122. The flow cytometry and adhesion assay data that support the findings of this study are available from the corresponding author upon reasonable request. Source data are provided with this paper.

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

## Acknowledgements

This work was funded by grants from the German Research Foundation (DFG, ZU377/4-1, Project-ID 375876048—TRR 241) and the Else Kröner Fresenius-Stiftung (2021_CS.23) to S.Z. The research of I.A., T.M.M., C.G., R.A., M.F.N. and S.Z. was further supported by the Interdisciplinary Center for Clinical Research (IZKF) and the ELAN program of the Universität Erlangen-Nürnberg, the Fritz-Bender-Stiftung, the Ernst Jung-Stiftung, the Else Kröner-Fresenius-Stiftung, the Thyssen-Stiftung, the German Crohn's and Colitis Foundation (DCCV), the DFG topic program on Microbiota, the Emerging Field Initiative, the DFG Collaborative Research Centers 643, 796, 1181 and TRR241 and the Clinical Research Unit 5024, the Rainin Foundation and the Litwin IBD Pioneers program of the Crohn's and Colitis Foundation of America (CCFA). The present work was performed in partial fulfillment of the requirements for obtaining the degree "PhD" for S.Z. The authors thank J. Derdau, D. Dziony, J. Marcks, J. Schuster and S. Hofmann for their invaluable technical assistance.

## Author contributions

MW performed the experiments. MW, LL and SZ designed the study and analyzed and interpreted the data. MD analyzed transcriptomic data together with MW and SZ. AS, EP, KAMU, LH, LW, FV, IA, TMM, CG, RA, MFN and SZ provided clinical samples, protocols or reagents; MW and SZ drafted the manuscript; all authors critically revised the manuscript for important intellectual content.

## FundingInformation

## Competing interests

M.F.N. has served as an advisor for Pentax, Giuliani, MSD, Abbvie, Janssen, Takeda and Boehringer. S.Z. received speaker's fees from Takeda, Roche, Galapagos, Ferring, Lilly, Falk and Janssen. M.F.N. and S.Z. received research support from Takeda, Shire (a part of Takeda) and Roche. The other authors declare no conflicts of interest.
