## [Peer Review File · Nature Communications]

Etrolizumab-s fails to control E-Cadherin-dependent co-stimulation of highly activated cytotoxic T cellsREVIEWER COMMENTS

Reviewer #1 (Remarks to the Author):

In this interesting study, the authors study the function of $\alpha\text{E}\beta 7$ on T cells in IBD, beyond cell trafficking. They also explore the mechanisms of action of etrolizumab and try to better understand the deceiving results with Etrolizumab in phase 3 trials. They show that cytotoxic T cells can be stimulated through interaction between $\alpha\text{E}\beta 7$ and E-cadherin expressed on epithelial cells. They do not explore how these cytotoxic mechanisms lead to uncontrolled inflammation. What are the hypothesis of the authors regarding the effect of these cytotoxic T cells? Do they think that these T cells could kill epithelial cells as shown in a recent publication (Front Immunol. 2022 Nov 10;13:1008456. doi: 10.3389)? In this study, Etrolizumab was able to reduce lympho-epithelial interactions and then cytotoxicity.

First, the authors describe a clear reduction of $\beta 7$ -expressing subsets in patients with active CD, probably related to their recruitment to the intestine. They show that a substantial portion of T cells expressing $\alpha 4\beta 7$ might be able to alternatively employ $\alpha 4\beta 1$ for gut homing in the presence of anti- $\alpha 4\beta 7$ antibodies. They performed adhesion assays with $\beta 7+$ memory T cells purified from the peripheral blood: cells were perfused through capillaries coated with MAdCAM-1 and VCAM-1 to mimic co-expression on the intestinal endothelium. Interestingly, a subset of $\beta 7+$ memory T cells co-expresses $\alpha 4\beta 1$, is still functional for gut homing even in the presence of vedolizumab. The authors used a cohort of IBD patients treated with vedolizumab to further explore if this might contribute to failure of $\alpha 4\beta 7$ blockade. They analyzed the % of peripheral blood $\alpha 4+\beta 7+\beta 1\text{hi}$ and $\alpha 4+\beta 7+\beta 1\text{lo}$ CD3+ T cells, and describe a selective reduction of $\alpha 4+\beta 7+\beta 1\text{hi}$ T cells in patients that did not enter in remission. Figure 3: the only result which is given is the Delta in % of $\alpha 4+\beta 7+\beta 1\text{hi}$, even if it's statistically significant the differences are not very striking. To be more convincing, I would suggest to provide more data on this with percentages at baseline and at the fifth infusion in patients in remission vs non-remission.

Interestingly, the authors further characterize these peripheral blood $\alpha 4+\beta 7+\beta 1\text{hi}$ CD3+ T cells using single cell RNA-sequencing, and show that they exhibit a cytotoxic signature. These transcriptome data are confirmed for protein levels (GLNY and GZMB) by FACS analysis. Then, the authors explored if memory T cells evading from vedolizumab effect could express CD103 in the tissue. TGF beta was still able to induce CD103 on these pro-inflammatory memory T cells. The authors then chose to focus on whether $\alpha\text{E}\beta 7$ - E Cadherin interactions drive T cell co-stimulation and how etrolizumab might interfere with this process. They used a co-stimulation assay with plate-coated E-Cadherin and anti CD3. S-Etro was able to reduce CD69 expression, as well as the secretion of Perforin, Granzyme B, soluble Fas ligand and Granulysin, at low concentration of anti-CD3 but not at high anti-CD3 concentration.

Comment: The authors consider that this effect of the high anti-CD3 concentration may reflect the in vivo situation of a highly stimulatory environment, which I think is overstated. Adding pro-inflammatory mediators or cytokines would be more appropriate to demonstrate this.

The authors had access to bulk RNA sequencing data from ileal samples of the cohort of the BERGAMOT phase III trial of etrolizumab in CD (34 patients treated with etrolizumab and 8 with placebo). Based on week 0 and 14 expression of five genes associated with T cell activation, they calculated a T cell activation score; and separated patients with a score (6 or higher), considered to have high T cell activation, to those with a score of 4 or lower. Interestingly, in those with a low T cell activation, Etrolizumab was associated with a reduction of cytotoxic mediators.

Comment: It would have been interesting to correlate these data with the response to Etrolizumab.

Reviewer #2 (Remarks to the Author):

Here the authors describe findings indicating that the gut homing of triple-integrin-expressing cells with the potential to evade blockade by vedolizumab might drive cytotoxicity and inflammation in intestinal tissues and that etrolizumab is able to reduce cytotoxic mediators in an environment with low, but not with high T cell stimulation.

MAJOR

1) Did the authors perform experiments (in vitro/functional) to assess whether etrolizumab might be able to reduce cytotoxic mediators in an environment with high T cell stimulation, when given at a higher dose?

2) Based on the findings described herein, can the authors speculate on whether CD103 would have been a better target than beta-7?

3) As we have come to appreciate for vedolizumab, the mechanism of action is not limited to the central compartment. Therefore, the lack of analysis of tissue biopsies of patients treated with etrolizumab is a major limitation of the current study.

MINOR

1) Authors should clearly indicate in the introduction the target of etrolizumab (i.e. beta-7 subunit found on alpha-4-beta-7 and alpha-E-beta-7).

Reviewer #3 (Remarks to the Author):

In this study, Wiendl et al. provide novel and important insights into the role of integrin-mediated gut homing of T cells. More importantly, their human data suggest potential mechanisms for the differential effects of Etrolizumab in clinical trials. It is worth noting that some of their mechanistic approaches raise questions. For instance, drawing conclusions on T cell activation and cytokine production based on bulk RNA sequencing data has obvious limitations. However, this caveat is being discussed by the authors and they provide a wide array of other assays (e.g. flow cytometry, in vitro experiments) that further support their findings. Overall, the quality of the findings is convincing, and the authors managed to generate impressive results.

However, as much as I am convinced about the significance of their findings and their methodological approach and results, I have some concerns regarding the structure of the manuscript as well as some of the conclusions drawn by the authors. Furthermore, I would recommend the authors to discuss unclear or even contradictory findings better. Finally, it appears to me as if part of the data is not related to the main topic of the manuscript, which results in confusion when reading the manuscript.

I outline my individual concerns in detail below:

1) The findings presented in Figure 2 and the related section of the manuscript are highly interesting. However, it is unclear how they relate to the remaining findings of the paper. The data presented in this figure suggest that the $\alpha 4\beta 1$ -VCAM-1 interaction might explain why certain $\beta 7+$ T cells do not respond to vedolizumab treatment in terms of their migration to the gut. It is worth noting that the main emphasis of this study was built around the potential mechanistic aspects of Etrolizumab treatment. Etrolizumab is not known to block $\alpha 4\beta 1$ -VCAM-1. After reading the introduction of the paper, I could not understand the rationale of studying this pathway and presenting it as a main figure. The fact that those findings do not relate to the rest of the manuscript is best highlighted by the fact that the results described in Figure 2 are not even mentioned in the discussion of the manuscript. In my opinion, those results should be shown as supplementary data or could be part of a separate manuscript. Including the data regarding $\alpha 4\beta 1$ -VCAM-1 in the current form makes the manuscript in my opinion difficult to follow. The authors could alternatively try to improve embedding and describing those findings in the overall context of the paper and how they relate to Etrolizumab.

2) In Figure 4 the authors show that $\alpha 4+\beta 7+\beta 1$ triple positive (expressing) cells are enriched in the peripheral blood of Crohn's disease compared to healthy control. The authors also described this in their corresponding results section. However, those results appear to be in contrast to the findings presented in Figure 1, where the authors have shown a reduced frequency of $\alpha 4+\beta 7+\beta 1$ hi cells in CD compared to control. This difference is likely based on the fact that the FACS data was generated in 14-27 patients, while the scRNA-seq data are based on a 1 vs. 1 comparison. However, the authors should discuss this difference when describing the results of the single-cell

sequencing data to prevent the readers from drawing false conclusions. I would recommend putting more emphasis on the fact that the scSeq data was used to identify a unique gene signature related to $\beta 7 + \beta 1 +$ T cells and not on the comparison of the frequency of $\beta 7 + \beta 1$ double expressing cells between healthy control and CD.

3) In the manuscript section related to Figure 4, the authors describe how the enrichment of $\beta 7 + \beta 1$ double expressing cells (which are actually $\alpha 4 \beta 7 \beta 1$ triple-positive since only $\alpha 4 +$ cells were sorted for sequencing) and their cytotoxic profile might promote intestinal inflammation.

Specifically, the authors conclude in the last sentences in this section of the manuscript:

“Consistent with the transcriptomic data, $\alpha 4 + \beta 7 + \beta 1 \text{hi}$ cells expressed higher levels of these mediators than $\alpha 4 + \beta 7 + \beta 1 \text{lo}$ cells and the difference was more pronounced in CD than in controls (Fig. 4H). Together, these findings indicated that the gut homing of triple-integrin-expressing cells with the potential to evade blockade by vedolizumab might drive cytotoxicity and inflammation in intestinal tissues.”

My main concern with the above-mentioned conclusion is the following: Figure 4 shows an increased presence of cytotoxic $\beta 7 + \beta 1$ double expressing cells in the blood of the IBD patient and the authors suggest that those cells promote intestinal inflammation in the setting of IBD. In Figure 1, however, the authors link increased frequencies of $\beta 7 + \beta 1$ double expressing cells in the peripheral blood to a mechanism that might actually be related to decreased intestinal inflammation.

Specifically, the following statement is made in the results section related to Figure 1:

“These data suggested that gut-homing T cells are depleted from the circulation probably due to recruitment to the intestine in CD and that a substantial portion of T cells expressing $\alpha 4 \beta 7$ might be able to alternatively employ $\alpha 4 \beta 1$ for gut homing in the presence of anti- $\alpha 4 \beta 7$ antibodies.”

Based on the data presented in Figure 4, one could also argue that the increased frequency of $\beta 7 + \beta 1$ double expressing cells with a cytotoxic profile in the peripheral blood might actually be a beneficial characteristic since those cells are not “depleted from circulation due to recruitment to the intestine in CD”.

Thus, my overall question is: do the authors interpret an increase of a certain population in the blood (such as $\beta 7 + \beta 1$ double expressing cells) as a potentially pathogenic signature (since those cells have cytotoxic properties) or as a beneficial signature (i.e. interpreting the increased frequencies in the blood as reduced migration of those cells into the intestine).

Overall, I believe that addressing and clarifying those issues would improve this work performed by Wiendl et al.

Point-by-point response:

We thank all the reviewers for taking their time to assess our manuscript and for making valuable suggestions to increase its quality. As detailed below, we have carefully addressed all the reviewers' comments in our revised manuscript.

Reviewer #1 (Remarks to the Author):

In this interesting study, the authors study the function of $\alpha E\beta 7$ on T cells in IBD, beyond cell trafficking. They also explore the mechanisms of action of etrolizumab and try to better understand the deceiving results with Etrolizumab in phase 3 trials. They show that cytotoxic T cells can be stimulated through interaction between $\alpha E\beta 7$ and E-cadherin expressed on epithelial cells. They do not explore how these cytotoxic mechanisms lead to uncontrolled inflammation. What are the hypothesis of the authors regarding the effect of these cytotoxic T cells? Do they think that these T cells could kill epithelial cells as shown in a recent publication (Front Immunol. 2022 Nov 10;13:1008456. doi: 10.3389)? In this study, Etrolizumab was able to reduce lympho-epithelial interactions and then cytotoxicity.

We thank the reviewer for raising this important point. Indeed, we think that these cytotoxic T cells can lead to epithelial cell death by release of cytotoxic mediators such as Fas ligand and Granzyme B. This is well established in the literature ^{1,2}. We highly valued the reference on the paper investigating T cell-epithelial interactions in organoid co-culture models. Similar to previous data from our own and other labs ^{3,4}, these data suggest that $\alpha E\beta 7$ mediates adhesive interactions between T cells and the epithelium, which also lead to T cells infiltrating the epithelium and functionally causes epithelial cell death ⁵ and that this can be blocked by etrolizumab. Our current data, however, promote the notion that co-stimulatory interactions are not sufficiently blocked by etrolizumab in a context of high T cell stimulation and, therefore, soluble cytotoxic mediators can still be active in a paracrine manner. We have added an additional paragraph to the discussion to reflect on these aspects.

First, the authors describe a clear reduction of $\beta 7$ -expressing subsets in patients with active CD, probably related to their recruitment to the intestine. They show that a substantial portion of T cells expressing $\alpha 4\beta 7$ might be able to alternatively employ $\alpha 4\beta 1$ for gut homing in the presence of anti- $\alpha 4\beta 7$ antibodies. They performed adhesion assays with $\beta 7+$ memory T cells purified from the peripheral blood: cells were perfused through capillaries coated with MAdCAM-1 and VCAM-1 to mimic co-expression on the intestinal endothelium. Interestingly, a subset of $\beta 7+$ memory T cells co-expresses $\alpha 4\beta 1$, is still functional for gut homing even in the presence of vedolizumab. The authors used a cohort of IBD patients treated with vedolizumab to further explore if this might contribute to failure of $\alpha 4\beta 7$ blockade. They analyzed the % of peripheral blood $\alpha 4+\beta 7+\beta 1hi$ and $\alpha 4+\beta 7+\beta 1lo$ CD3+ T cells, and describe a selective reduction of $\alpha 4+\beta 7+\beta 1hi$ T cells in patients that did not enter in remission. Figure 3: the only result which is given is the Delta in % of $\alpha 4+\beta 7+\beta 1hi$, even if it's statistically significant the differences are not very striking. To be more convincing, I would suggest to provide more data on this with percentages at baseline and at the fifth infusion in patients in remission vs non-remission.

As requested by the reviewer, we have added graphs showing the percentages at baseline and at treatment five to Figure 3. These data further support our conclusion.

Interestingly, the authors further characterize these peripheral blood $\alpha 4+\beta 7+\beta 1hi$ CD3+ T cells using single cell RNA-sequencing, and show that they exhibit a cytotoxic signature. These transcriptome data are confirmed for protein levels (GLNY and GZMB) by FACS analysis.

Then, the authors explored if memory T cells evading from vedolizumab effect could express CD103 in the tissue. TGF beta was still able to induce CD103 on these pro-inflammatory memory T cells. The authors then chose to focus on whether $\alpha E\beta 7$ - E Cadherin interactions drive T cell co-stimulation and how etrolizumab might interfere with this process. They used a co-stimulation assay with plate-coated E-Cadherin and anti CD3. S-Etro was able to reduce CD69 expression, as well as the secretion of Perforin, Granzyme B, soluble Fas ligand and Granulysin, at low concentration of anti-CD3 but not at high anti-CD3 concentration.

Comment: The authors consider that this effect of the high anti-CD3 concentration may reflect the in vivo situation of a highly stimulatory environment, which I think is overstated. Adding pro-inflammatory mediators or cytokines would be more appropriate to demonstrate this.

We agree with the reviewer that the conditions in our in vitro assays do not adequately reflect a "highly stimulatory environment" in the inflamed gut in vivo. Our intention was to characterize the level of T cell receptor stimulation in analogy to our experiments. Since we aimed at investigating a co-stimulatory role of $\alpha E\beta 7$ integrin during TCR stimulation, we are convinced that adding pro-inflammatory mediators or cytokines instead of anti-CD3 would not have been the setting to answer our question. However, we have carefully rephrased the manuscript including the title to avoid any overstatement regarding the conclusions to be drawn from high anti-CD3 concentrations for the in vivo situation.

The authors had access to bulk RNA sequencing data from ileal samples of the cohort of the BERGAMOT phase III trial of etrolizumab in CD (34 patients treated with etrolizumab and 8 with placebo). Based on week 0 and 14 expression of five genes associated with T cell activation, they calculated a T cell activation score; and separated patients with a score (6 or higher), considered to have high T cell activation, to those with a score of 4 or lower. Interestingly, in those with a low T cell activation, Etrolizumab was associated with a reduction of cytotoxic mediators.

Comment: It would have been interesting to correlate these data with the response to Etrolizumab.

We completely agree with the reviewer that it would have been very informative to correlate these data with the response to etrolizumab. Unfortunately, however, we do only have access to the RNA seq data and not to the corresponding clinical data and can therefore not determine response or non-response in these patients. We have mentioned this limitation in the revised manuscript.

Reviewer #2 (Remarks to the Author):

Here the authors describe findings indicating that the gut homing of triple-integrin-expressing cells with the potential to evade blockade by vedolizumab might drive cytotoxicity and inflammation in intestinal tissues and that etrolizumab is able to reduce cytotoxic mediators in an environment with low, but not with high T cell stimulation.

MAJOR

1) Did the authors perform experiments (in vitro/functional) to assess whether etrolizumab might be able to reduce cytotoxic mediators in an environment with high T cell stimulation, when given at a higher dose?

We thank the reviewer for this excellent suggestion. We have performed additional co-stimulation cultures to address this question. Accordingly, CD8⁺ T cells were prepared in analogy to the previously described assays and stimulated with anti-CD3 +/- E-Cadherin. Etolizumab-s was used at concentrations of 10µg/ml and 100µg/ml and we quantified the secretion of cytotoxic mediators to the culture supernatant. Indeed, 100µg/ml were able to modestly, but significantly reduce the production of these mediators. Thus, very high (supraphysiological concentrations) of etolizumab seem to be able to compensate for the lack of efficacy of physiological etolizumab-s levels to interfere with αEβ7-driven T cell co-stimulation at high levels of T cell stimulation. We have included these data as a new Supplementary Figure 6.

2) Based on the findings described herein, can the authors speculate on whether CD103 would have been a better target than beta-7?

Our data suggest two potential answers to this very interesting questions: i) Either, indeed, CD103 is the better target than beta-7, since we observed that the ability of anti-CD103 to block co-stimulation was preserved even at high levels of T cell stimulation, or ii) the limited efficacy of etolizumab in this regard is an antibody-specific phenomenon and other antibodies blocking beta-7 at different epitopes might overcome this problem. As requested, we have speculated on this aspect in the revised discussion of the manuscript.

3) As we have come to appreciate for vedolizumab, the mechanism of action is not limited to the central compartment. Therefore, the lack of analysis of tissue biopsies of patients treated with etolizumab is a major limitation of the current study.

We fully agree with the reviewer. We would have very much liked to investigate our concept in tissue biopsies from patients treated with etolizumab, but this is obviously not possible, when a compound is not clinically available. We had already clearly stated this as a limitation of our study and have now further underscored this point in the revised discussion to reflect the reviewer's concern.

MINOR

1) Authors should clearly indicate in the introduction the target of etolizumab (i.e. beta-7 subunit found on alpha-4-beta-7 and alpha-E-beta-7).

As requested, we have clearly indicated this in the introduction.

Reviewer #3 (Remarks to the Author):

In this study, Wiendl et al. provide novel and important insights into the role of integrin-mediated gut homing of T cells. More importantly, their human data suggest potential mechanisms for the differential effects of Etolizumab in clinical trials. It is worth noting that some of their mechanistic approaches raise questions. For instance, drawing conclusions on T cell activation and cytokine production based on bulk RNA sequencing data has obvious limitations. However, this caveat is being discussed by the authors and they provide a wide array of other assays (e.g. flow cytometry, in vitro

experiments) that further support their findings. Overall, the quality of the findings is convincing, and the authors managed to generate impressive results.

We thank the reviewer for his/her assessment of our study.

However, as much as I am convinced about the significance of their findings and their methodological approach and results, I have some concerns regarding the structure of the manuscript as well as some of the conclusions drawn by the authors. Furthermore, I would recommend the authors to discuss unclear or even contradictory findings better. Finally, it appears to me as if part of the data is not related to the main topic of the manuscript, which results in confusion when reading the manuscript.

I outline my individual concerns in detail below:

1) The findings presented in Figure 2 and the related section of the manuscript are highly interesting. However, it is unclear how they relate to the remaining findings of the paper. The data presented in this figure suggest that the $\alpha 4\beta 1$ -VCAM-1 interaction might explain why certain $\beta 7+$ T cells do not respond to vedolizumab treatment in terms of their migration to the gut. It is worth noting that the main emphasis of this study was built around the potential mechanistic aspects of Etrolizumab treatment. Etrolizumab is not known to block $\alpha 4\beta 1$ -VCAM-1. After reading the introduction of the paper, I could not understand the rationale of studying this pathway and presenting it as a main figure. The fact that those findings do not relate to the rest of the manuscript is best highlighted by the fact that the results described in Figure 2 are not even mentioned in the discussion of the manuscript. In my opinion, those results should be shown as supplementary data or could be part of a separate manuscript. Including the data regarding $\alpha 4\beta 1$ -VCAM-1 in the current form makes the manuscript in my opinion difficult to follow. The authors could alternatively try to improve embedding and describing those findings in the overall context of the paper and how they relate to Etrolizumab.

We thank the reviewer for these valuable suggestions to improve the narrative of our story. We apologize for not making the connection of the $\alpha 4\beta 1$ -VCAM-1 aspect with etrolizumab clear enough. The idea was to interrogate the rationale to target $\alpha E\beta 7$ in addition to $\alpha 4\beta 7$ and, in the first part of the manuscript, we therefore investigated whether $b 7$ -expressing T cells that co-express $a 4b 1$ can bypass vedolizumab therapy via this pathway, may subsequently contribute to pro-inflammatory signaling in the gut and may express $\alpha E\beta 7$ to become potential targets of etrolizumab. We have substantially re-arranged the manuscript to connect this aspect better to the second part exploring the role of $\alpha E\beta 7$ for T cell co-stimulation and the potential of etrolizumab to block it:

- *We have revised the introduction to provide a clear rationale for studying the $\alpha 4\beta 1$ -VCAM-1 pathway.*
- *We also mention the data from former Figure 2 in the revised discussion.*
- *As suggested, we have moved Figure 2 to the Supplement (now part of Supplementary Figure 2 and 3).*
- *We have improved the embedding of these findings into the narrative of the results section.*

2) In Figure 4 the authors show that $\alpha 4+\beta 7+\beta 1$ triple positive (expressing) cells are enriched in the peripheral blood of Crohn's disease compared to healthy control. The authors also described this in their corresponding results section. However, those results appear to be in contrast to the findings presented in Figure 1, where the authors have shown a reduced frequency of $\alpha 4+\beta 7+\beta 1$ hi cells in CD compared to control. This difference is likely based on the fact that the FACS data was generated in

14-27 patients, while the scRNA-seq data are based on a 1 vs. 1 comparison. However, the authors should discuss this difference when describing the results of the single-cell sequencing data to prevent the readers from drawing false conclusions. I would recommend putting more emphasis on the fact that the scSeq data was used to identify a unique gene signature related to $\beta7+\beta1+$ T cells and not on the comparison of the frequency of $\beta7+\beta1$ double expressing cells between healthy control and CD.

We agree with the reviewer that the previous presentation of the data in Figure 4 may create confusion with regard to the frequency of $\alpha4+\beta7+\beta1+$ cells in CD compared to healthy control.

It is important to note that a direct comparison of the flow cytometry data shown in Figure 1 with the single cell sequencing data in Figure 4 (now Figure 3) is not possible. In Figure 1, peripheral blood mononuclear cells were analyzed and gated on T cells with all the frequencies indicated relating to CD4+ or CD8+ memory T cells. For the transcriptome analysis, $\alpha4$ integrin+ memory T cells were sorted and used in approximately equal numbers. Thus, it is not possible to relate cell numbers to total memory T cells per patient. Moreover, the assessment of mRNA vs. protein level and technical aspects such as dropouts also preclude a direct comparison.

However, to avoid any ambiguity, we followed the reviewer's recommendation to put more emphasis on the identification of gene signatures and not on the comparison of cell frequencies. Accordingly, we have modified the previous Figure 4 (now Figure 3) and the associated paragraphs in the text.

3) In the manuscript section related to Figure 4, the authors describe how the enrichment of $\beta7+\beta1$ double expressing cells (which are actually $\alpha4 \beta7 \beta1$ triple-positive since only $\alpha4+$ cells were sorted for sequencing) and their cytotoxic profile might promote intestinal inflammation. Specifically, the authors conclude in the last sentences in this section of the manuscript:

“Consistent with the transcriptomic data, $\alpha4+\beta7+\beta1$ hi cells expressed higher levels of these mediators than $\alpha4+\beta7+\beta1$ lo cells and the difference was more pronounced in CD than in controls (Fig. 4H). Together, these findings indicated that the gut homing of triple-integrin-expressing cells with the potential to evade blockade by vedolizumab might drive cytotoxicity and inflammation in intestinal tissues.”

My main concern with the above-mentioned conclusion is the following: Figure 4 shows an increased presence of cytotoxic $\beta7+\beta1$ double expressing cells in the blood of the IBD patient and the authors suggest that those cells promote intestinal inflammation in the setting of IBD. In Figure 1, however, the authors link increased frequencies of $\beta7+\beta1$ double expressing cells in the peripheral blood to a mechanism that might actually be related to decreased intestinal inflammation.

Specifically, the following statement is made in the results section related to Figure 1:

“These data suggested that gut-homing T cells are depleted from the circulation probably due to recruitment to the intestine in CD and that a substantial portion of T cells expressing $\alpha4\beta7$ might be able to alternatively employ $\alpha4\beta1$ for gut homing in the presence of anti- $\alpha4\beta7$ antibodies.”

Based on the data presented in Figure 4, one could also argue that the increased frequency of $\beta7+\beta1$ double expressing cells with a cytotoxic profile in the peripheral blood might actually be a beneficial characteristic since those cells are not “depleted from circulation due to recruitment to the intestine in CD”.

Thus, my overall question is: do the authors interpret an increase of a certain population in the blood (such as $\beta7+\beta1$ double expressing cells) as a potentially pathogenic signature (since those cells have cytotoxic properties) or as a beneficial signature (i.e. interpreting the increased frequencies in the blood as reduced migration of those cells into the intestine).

We thank the reviewer for this valuable comment, which is closely related to the previous point.

We and others have repeatedly observed (and functionally examined) that changes in the frequency of cell subsets in the peripheral blood are mirrored by reverse changes in the gut in the context of cell trafficking in IBD⁶⁻⁹. We therefore completely stand to the conclusion presented along with description of the data in Figure 1.

With regard to Figure 4, we apologize for any confusion that may have arisen from the presentation of the frequencies of ITGB7+ITGB1+ cells from the patient with CD and the control donor. As mentioned above, there are several aspects that preclude a direct comparison of these frequencies with the frequencies as determined and shown in Figure 1. Thus, it was not at all our intention to draw any conclusions about the frequencies of these cells in the peripheral blood based on single cell sequencing, but rather to characterize their distribution throughout the single cell transcriptomic dataset. According to the reviewer's previous comment we have therefore modified Figure 4 (now Figure 3) to focus on gene expression signatures and not on cell frequencies.

These data show that triple integrin expressing cells have a particularly cytotoxic phenotype. Therefore, our overall concept is that triple-integrin-expressing cells seem to be depleted from the peripheral circulation due to gut homing in CD and that due to their cytotoxic phenotype they might serve pro-inflammatory functions in intestinal tissues. Hence, we also stand to the cited conclusion derived from the Figure 4 data (now Figure 3). We are confident that the modified presentation of the data in Figure 4 (now Figure 3) according to the valuable suggestions of the reviewer will help readers to comprehend our reasoning and will prevent any confusion.

Overall, I believe that addressing and clarifying those issues would improve this work performed by Wiendl et al.

Once again, we thank the reviewer for his/her critique and believe that our modifications along the reviewers' guidance have substantially improved our work.

References

1. Lin, T. *et al.* Fas ligand- mediated killing by intestinal intraepithelial lymphocytes. Participation in intestinal graft-versus-host disease. *J Clin Invest* **101**, 570–577 (1998).
2. Cupi, M. L. *et al.* Plasma Cells in the Mucosa of Patients with Inflammatory Bowel Disease Produce Granzyme B and Possess Cytotoxic Activities. *The Journal of Immunology* **192**, 6083–6091 (2014).
3. Zundler, S. *et al.* Blockade of $\alpha\text{E}\beta 7$ integrin suppresses accumulation of CD8+ and Th9 lymphocytes from patients with IBD in the inflamed gut in vivo. *Gut* **66**, 1936–1948 (2017).
4. Dai, B. *et al.* Dual targeting of lymphocyte homing and retention through $\alpha 4\beta 7$ and $\alpha\text{E}\beta 7$ inhibition in inflammatory bowel disease. *Cell Rep Med* **2**, 100381 (2021).
5. Hammoudi, N. *et al.* Autologous organoid co-culture model reveals T cell-driven epithelial cell death in Crohn's Disease. *Frontiers in Immunology* **13**, (2022).
6. Vermeire, S. *et al.* Etrolizumab as induction therapy for ulcerative colitis: a randomised, controlled, phase 2 trial. *The Lancet* **384**, 309–318 (2014).
7. Zundler, S. *et al.* The $\alpha 4\beta 1$ Homing Pathway Is Essential for Ileal Homing of Crohn's Disease Effector T Cells In Vivo. *Inflamm. Bowel Dis.* **23**, 379–391 (2017).
8. Schleier, L. *et al.* Non-classical monocyte homing to the gut via $\alpha 4\beta 7$ integrin mediates macrophage-dependent intestinal wound healing. *Gut* **69**, 252–263 (2020).
9. Binder, M.-T. *et al.* Similar Inhibition of Dynamic Adhesion of Lymphocytes From IBD Patients to MAdCAM-1 by Vedolizumab and Etrolizumab-s. *Inflamm. Bowel Dis.* **24**, 1237–1250 (2018).

REVIEWERS' COMMENTS

Reviewer #1 (Remarks to the Author):

The manuscript has been clearly improved since the first submission. I consider that authors responded to all my comments and modified the manuscript accordingly.

Reviewer #2 (Remarks to the Author):

Thank you for addressing the reviewer concerns.

Reviewer #3 (Remarks to the Author):

The authors have addressed my concerns and have improved their manuscript. Thus, I recommend the publication of the manuscript in its current form.